# Microstructure and Mechanical Properties of V-Alloyed Rebars Subjected to Tempcore Process

**Essam Ahmed** [1,*] , **Samir Ibrahim** [1] , **Mohamed Galal** [2] , **Sarah A. Elnekhaily** [1] **and Tarek Allam** [1,3,*]

[1] Department of Metallurgical and Materials Engineering, Faculty of Petroleum and Mining Engineering, Suez University, Suez 43512, Egypt; saibrahim158@gmail.com (S.I.); s.elnekhaily@suezuni.edu.eg (S.A.E.)
[2] Al Ezz Dekheila Steel Company (EZDK), El-Dekheila, Alexandria 21537, Egypt; mogalal@ezzsteel.com.eg
[3] Steel Institute (IEHK), RWTH Aachen University, D-52056 Aachen, Germany
[*] Correspondence: essam.ahmed@suezuniv.edu.eg (E.A.); tarek.allam@iehk.rwth-aachen.de (T.A.)

**Abstract:** Two B400B-R and B500B grade rebars were industrially produced through a Tempcore process. The standard chemical composition of B500B grade was additionally alloyed with 0.067 wt.% V to enhance its mechanical properties. A set of optimized processing parameters were applied to manufacture two different diameters D20 (Ø 20 mm) and D32 (Ø 32 mm). The microstructure -mechanical properties relationships were evaluated using optical and scanning electron microscopes, hardness, and tensile testing. In addition, a thermal model was developed to define the thermal cycle evolution during cooling in the quenching & tempering box (QTB) to simulate the kinetics of V(C,N) precipitation. The microstructure observations showed a typical graded microstructure consisting of ferrite-pearlite core and outer tempered martensite ring for both grades of both diameters. The optimized processing parameters for B400B-R of D32 (compared with D20) resulted in softening of the core (from 160 to 135 HV10) and tempered martensite surface (from 220 to 200 HV10) as well as in decreasing the yield strength (from 455 to 413 MPa) and tensile strength (from 580 to 559 MPa). On the contrary, an increase in hardness of the core (from 165 to 175 HV10) and the outer tempered martensite (from 240 to 270 HV10), in addition to an increase in yield strength (from 510 to 537 MPa) at almost the same level of tensile strength of 624–626 MPa are observed for B500B grade D32 compared with D20. The modeling and simulation calculations suggest that the manufacturing D32 rebars of B500B grade involves longer quenching time in the QTB which allow deeper tempered martensite surface along with a relatively higher core temperature that renders faster kinetics and larger volume fraction of V(C,N) precipitates. The current study demonstrates that the full potential of V-alloying can be exploited when a sufficient quenching time at the equalization temperature is achieved, which is valid for D32 rebars.

**Keywords:** Tempcore processing; quenching & tempering box (QTB); V-alloyed steel; V(C,N) precipitations; microstructure; mechanical properties

## 1. Introduction

Searching for more energy efficient solutions has led to an increased interest in the development of structural steels using cost effective production processes [1,2]. The quenching and self-tempering (QST) or Tempcore process has been widely applied in the production of low C-Mn steel rebars due to its relatively low cost compared to processes based on microalloying addition or conventional cold working [3–5]. The Tempcore technique allows the production of concrete reinforcing bars with high mechanical properties alongside excellent weldability and superior ductility and bendability. This technique can guarantee excellent process controllability and high flexibility. However, it requires reducing the rolling line speed for efficient cooling to achieve the required mechanical properties, compared with that of the ordinary hot rolling. Thus, a high quenching time for the steel rebars, inside the cooling box, will be on the expense of in-line productivity. The Tempcore process can be summarized as follows: following the hot-rolling stage, at ~1000 °C, the hot rolled rebar

enters the cooling zone in which it is quenched. Quenching is done using water sprays that rapidly cool the surface of the bar below the martensite start (Ms) temperature [6]. Thus, a hardened surface layer "case" is created while the central zone/core of the bar remains austenitic until it transforms during further air cooling. Between the surface and core zones a transition zone develops, where the cooling rate could be sufficient to trigger lower or upper bainitic phase transformation depending on the applied process parameters. In a subsequent step, heat transfers from the core towards the outer surface rim, tempering the martensite [7–9] and hence, developing a self-tempering process. Tempering results in stress relief due to the diffusion of carbon out of the martensite depending on the carbon content. Within this step, the surface of the bar is reheated to approximately 600–700 °C (the equalization temperature) and subsequently cooled naturally until reaching ambient temperatures on the cooling bed. The Tempcore process route is distinguished from the conventional normalizing hot-rolling route, where the rebars leave the finishing rolling mill and cool down directly in air [10]. The major difference between the two processes lies in the final temperature before air cooling of rebars. The Tempcore process allows lower final temperatures compared to normalizing rolling route resulting in a graded microstructure consisting of outer hard self-tempered martensite case and soft ferrite-pearlite core. The core is softer and more ductile than that of the other fully micro-alloyed counterparts, which can be achieved by the reduction of alloying additions [11]. The Tempcore process that take place in the quenching and tempering box (QTB) has many input variables that must be kept under control to obtain the desired properties. These input variables are the finishing temperature, number of coolers, number of strippers, water pressure, water flow, finishing speed and cooler size relative to the bar size. The most important output variable is achieving the desired mechanical properties [12,13]. Assuming the same chemistry, the primary determining factor of the strength of the bar is the quench depth or the tempered zone. As such, measuring the tensile properties of the bar gives a very good indication of the quenching systems. Microalloying elements, such as vanadium, are often added in the steel rebar to enhance mechanical properties through grain refinement of ferrite and nanoscale precipitates strengthening [14–16]. Weldability criteria are used to estimate allowable carbon equivalent values according to Equation (1) [17] (Ceq. = 0.35 and 0.45), i.e., values at which no cold cracks or other welding defects are formed. The alloying elements content in steels of the C–Mn–Si–V system are optimized for the assembly of welded reinforcing cages [18]. So far, a systematic investigation to explore the extent of exploitation of the full potential of V-alloying during manufacturing different rebar diameters is missing. Moreover, the influence of temperature evolution throughout the whole diameter of rebars on the kinetics of V(C,N) precipitation to tailor the corresponding microstructure and the mechanical properties is not fully understood yet. In the present work, a set of optimized Tempcore processing parameters are applied to manufacture two different rebar steel grades (B400B-R and V-alloyed B500B) with different diameters, namely, D20 (20 mm) and D32 (32 mm). In addition, a finite element model is developed to predict the evolution of temperature profile during cooling in the QTB and after air-cooling, which is used to calculate the precipitation kinetics of V(C,N). The differences in microstructure and mechanical properties due to manufacturing different of the examined rebars grades are investigated and discussed and correlated with the potential of V(C,N) precipitations for B500B grade.

## 2. Materials and Methods

The investigated two steel variants B400B-R (containing 0.003 wt.% V) and B500B (containing 0.067 wt.% V) were industrially manufactured on a mass-production scale at the bar mill of Ezz Dekheila Steel Co. (EZDK, Alexandria, Egypt). The target chemical compositions of B400B-R and B500B were adjusted in the ladle furnace according to Egyptian and British standards, respectively. Table 1 lists the measured chemistries of the corresponding produced steel bars using an optical emission spectroscopy (DP73, Olympus, Tokyo, Japan). The continuously casted billets with a cross-section of 130 mm × 130 mm

are further thermo-mechanically processed in 16 stands bar hot rolling-mill before they experience a controlled cooling in the QTB. The typical operation parameters used for rebars with 20 and 32 mm diameters of different steel compositions are summarized in Table 2.

$$Ceq = \%C + \frac{\%Mn}{6} + \frac{\%Cu}{40} + \frac{\%Ni}{20} + \frac{\%Cr}{10} - \frac{\%V}{10} - \frac{\%Mo}{50} \tag{1}$$

**Table 1.** Chemical composition of used steels rebars.

| Steel Grade | Chemical Analysis, wt.% | | | | | | | | | | |
|---|---|---|---|---|---|---|---|---|---|---|---|
| | C | Si | Mn | P | S | Cr | Ni | V | Cu | N | Ceq. |
| B400B-R | 0.28 | 0.14 | 0.73 | 0.015 | 0.025 | 0.05 | 0.07 | 0.003 | 0.29 | 0.0068 | 0.42 |
| B500B | 0.22 | 0.17 | 1.37 | 0.017 | 0.025 | 0.05 | 0.08 | 0.067 | 0.15 | 0.0057 | 0.45 |

**Table 2.** Typical operation parameters for the QTB rebars production.

| Bar Diameter (mm) | QTB Normal Rolling Speed (m/s) | Actual Hot Rolling Speed (m/s) | Achieved Rolling Speed with QTB * (m/s) | Water Flow (m$^3$/h) | Equalization Temperature (°C) |
|---|---|---|---|---|---|
| 20 | 8 | 13 | 13 | 600 | 665 |
| 32 | 4 | 7 | 7 | 620 | 670 |

* Nearly the same rolling speed is achieved with modified QTB when compared to ordinary hot rolling (no reduction in productivity). The finishing rolling temperature, number of cooling pipes, and water pressure are fixed at 1000 °C, 10, and 12 Bar, respectively.

The microstructure characteristics of the different developed zones were observed for both steel grades using an optical microscope (DP73, Olympus, Tokyo, Japan) as well as a Quanta FEG 250 scanning electron microscope, FEI company (Hillsboro, OR, USA). The standard sample preparation procedures were applied. The metallographic samples were ground progressively using wet silicon carbide emery papers with grit number starting with 180-grit and proceeding to 240-, 400-, 500-, 600-, 800-, 1000- and 1200-grit papers, and finally polished on a low-speed wheel covered with micro-cloth using 0.05 μm Al$_2$O$_3$ suspension. To reveal the microstructure, the samples were etched using nital reagent for a period of 4–6 s at room temperature. The line intercept method was applied to measure the average grain size according to the guidelines of ASTM E-112 standard.

The case depths and thus case areas are measured using a micrometer fitted with the microscope. The mechanical properties: yield strength (*YS*), ultimate tensile strength (*UTS*), and total elongation percentage (El%) of the steel rebars were evaluated by means of quasi-static tensile testing. The tensile tests were conducted using a universal tensile testing machine Instron 4210 at room temperature with a cross-head speed of 0.1 mm/s on cylindrical specimens of 100, and 160 mm parallel gauge length for D20 and D32, respectively, according to the DIN EN10002-1 2001 standard. Vickers hardness measurements were carried out to visualize the hardness distribution over the microstructurally distinguished zones with applying a test load of 10 kgf and dwell time of 15 s. The mechanical properties were evaluated based on average values calculated from at least three successfully tested samples.

A thermal model is developed to predict the heat transfer process and the temperature distribution for both the quenching and the subsequent cooling processes on reaching the rebar equalization temperature. The temperature distribution is predicted by finite element analysis using ABAQUS (version 6.14.1, Dassault Systèmes Simulia Corp., Providence, RI, USA) software. The model is conducted on two-dimensional uniform semi circles. The heat transfer is assumed to be negligible along the rebar length. Only the radial flow across the cross section is considered [19]. The quenching and self-tempering processes are carried out through a conduction/convection heat transfer transient problem, which can

be summarized into three main steps. The first step is to calculate the film coefficient ($h$) between the rebar and water during quenching. The film coefficient depends mainly on temperature among other variables such as: rebar diameter, tube diameter, water flow rate, and water pressure [20–22]. For simplicity, the coefficient could be taken as an average fixed value, for each rebar diameter, throughout the model. Several studies have managed to estimate this value after intensive numerical analysis that relates the heat transfer coefficient to the cooling process parameters [23,24]. The analysis is based on adjusting the film coefficient to obtain temperatures that are equal or nearly equal to that measured at the inlet and outlet of the Tempcore box. Dimatteo et al. [22], established an equation that predicts $h$ from the rebar geometry and of the cooling water flow, Equation (2):

$$h = a_1 + a_2 \left(D + D'\right) + a_3 \frac{D}{D'} + a_4 \, x \, W \tag{2}$$

where $D$ and $D'$ are the rebar and the cooling tube diameters, respectively. $\dot{W}$ is the water flow rate in the cooling tube, and $a_1$, $a_2$, $a_3$, and $a_4$ are the equation constant parameters. Bandyopadhyay et al. [23] found that the heat transfer coefficients of water in the Tempcore box with values 15 and 40 kW/m$^2$ K for the 16 and 32 mm steel rebar diameters, respectively, gave adequate results. In the present work, the film coefficients are estimated for each rebar diameter from [23,24] by which a good agreement is shown between measured and calculated temperatures as illustrated in the following section. The second step is the water quenching process, modeled numerically, by which temperature distribution after quenching is predicted. A uniform initial temperature of 1000 °C is applied to the steel rebar. The temperature distribution across the rebar cross-section with time is given in the following equation:

$$\frac{\partial}{\partial r}\left(k\frac{\partial T}{\partial r}\right) + \frac{k}{r}\left(\frac{\partial T}{\partial r}\right) = \rho C_p \frac{\partial T}{\partial t} \tag{3}$$

where, $T$ is the temperature, $r$ is the distance from cold surface, $k$ is the thermal conductivity, $Cp$ is the specific heat, $\rho$ is the steel density and $t$ is the time [25]. The steel thermal conductivity ($k$) and the specific heat ($Cp$) are added to the model as functions of temperature [26,27]. The boundary conditions adopted in the thermal model are as follows:

At the rebar cross section center:

$$\frac{\partial T}{\partial r}_{r=0} = 0 \tag{4}$$

At the rebar surface:

$$- k\frac{\partial T}{\partial r}_{r=R} = h[T(R,t) - T_\infty] \tag{5}$$

where, $T_\infty$ is the temperature of the surrounding medium (water) which is taken here as 25 °C (298 K). The final step is self-tempering where the equalization temperature is reached during an air-cooling process. The film coefficient in this step has a relatively small value which could be taken as 40 W/m$^2$ K [24]. It is worth mentioning that the numerical model, in this work, is concerned only with the heat treatment process the rebar experiences, while the thermodynamic equilibrium phase evolution was calculated using Thermo-Calc software TCFE Steels/Fe-alloys database version 10 (Thermo-Calc Software, Stochholm, Sweden). The main aim of this model is to determine the temperature distribution and to predict the thermal cycles (the minimum and equalization temperatures) achieved, which were applied to simulate the precipitation kinetics of V(C,N) using MatCalc software version 6.03 (MatCalc Engineering GmbH, Vienna, Austria). The thermal analysis is performed with input data related to processing parameters used in the present steel rebar production. The thermal model results are validated with experimental ones.

## 3. Results and Discussion

### 3.1. Thermo Dynamic Calculations

The equilibrium phase evolutions of the investigated steels are represented in Figure 1. Obviously, the B500B steel grade containing 0.067 wt.% V shows the formation of FCC_A1#2 phase almost below 1000 °C referring to the start of precipitation of V(C,N) particles in the austenite phase and extends into ferrite phase as well under the equilibrium conditions (Figure 1a). On the contrary, the phase evolution diagram of B400B-R steel grade does not indicate the formation of FCC_A1#2 phase (Figure 1b). The equilibrium ferrite start temperature for both steel grades is around 800 °C, however, the cementite starts to form when the temperature firstly drops to approximately 700 °C, below which the austenite completely decomposes.

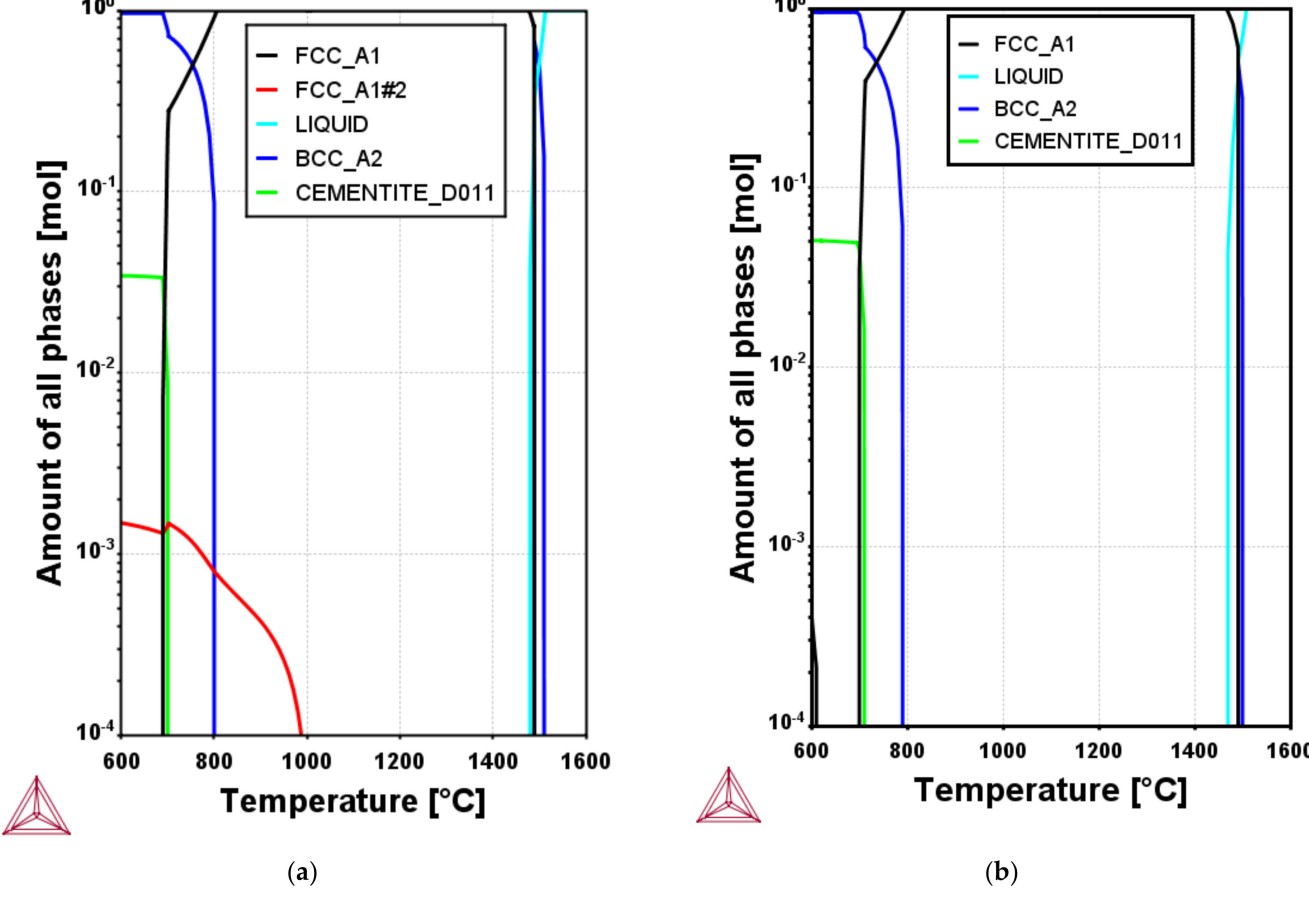

(a)                                                                                          (b)

**Figure 1.** Phase evolutions of the investigated steel grades calculated using ThermoCalc software (TCFE 10 database). The equilibrium amount of all phases vs. temperature for B500B and B400B-R steel grades are shown in (**a**) and (**b**), respectively. FCC_A1, FCC_A1#2 and BCC_A2 phases stand for austenite, V(C,N) and ferrite, respectively.

### 3.2. Microstructure Characteristics

#### 3.2.1. Outer Surface (rim)

From microstructural examination, a tempered martensite with lath like morphology is observed. Figure 2 shows that the grain structure of martensite is consisting of differently oriented martensite packets containing various blocks of martensite lathes. Such typical martensitic microstructure with three level-hierarchy i.e., packets, blocks and lathes were reported by Krauss [28]. The pronounced variation in orientations of the developed martensitic microstructure emerges from the several possible crystallographic orientation relationships between martensite and the parent phase austenite. Although the Kurdjumov-Sachs (K-S) orientation relationship model considers 24 variants to form martensite from

austenite grains and 6 possible variants in each packet. Kitahara et al. [29] pointed out that neither all the 24 austenite grain variants nor the 6 packet variants can essentially appear. A narrower lath is obtained for the smaller 20 mm rebar as shown in Figure 2a,c attributable to the relatively lower equalization temperature achieved after during process. The SEM micrographs of the outer layer of the B500B steel grade shown in Figure 3a,b indicate the possible carbonitride precipitates during the Tempcore process. The boundaries of martensite packets seem to be decorated with fine carbonitride precipitates, which could not be resolved at the applied magnification.

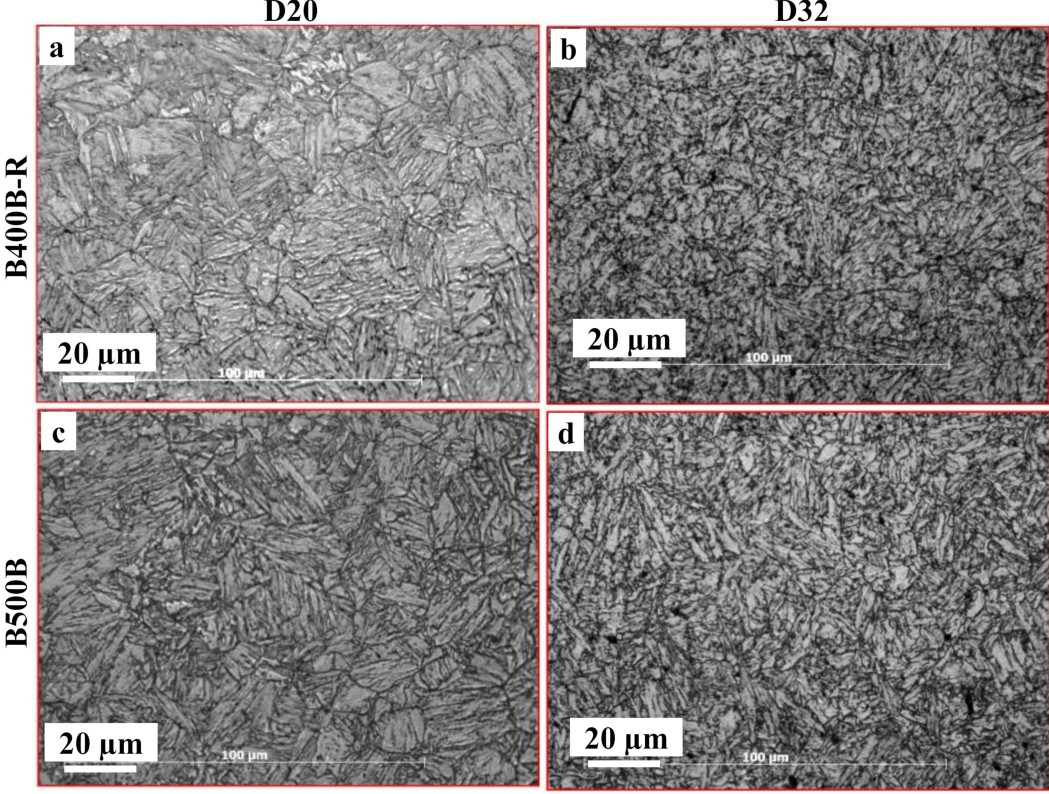

**Figure 2.** Optical micrographs showing the developed microstructures in the outer surfaces (**a**,**b**) and (**c**,**d**) of the D20 and D32 B400B-R and B500B steel grades, respectively.

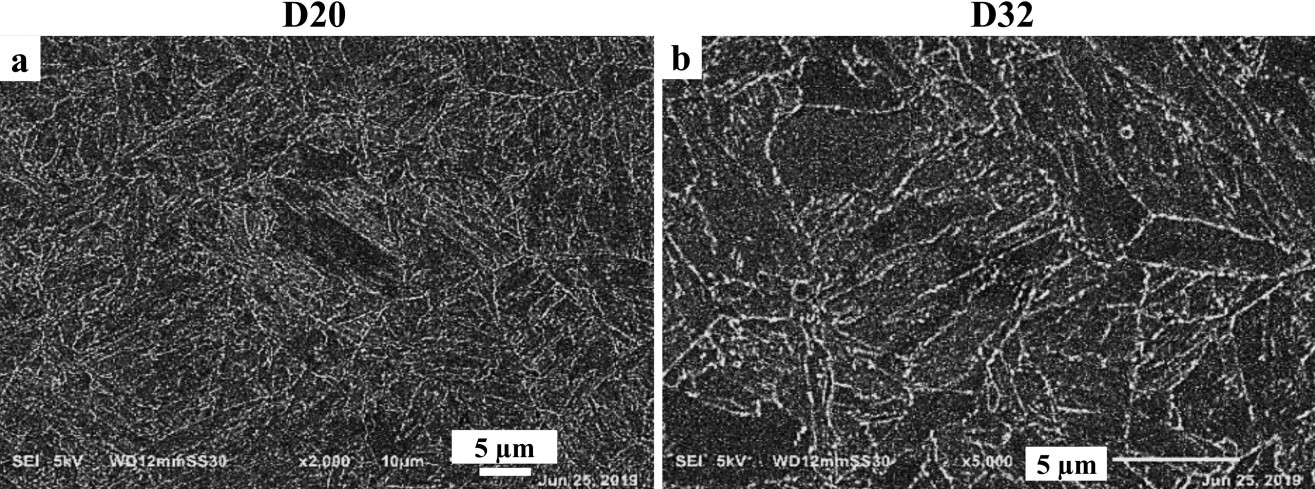

**Figure 3.** SEM micrographs of B500B steel grade. (**a**,**b**) show the tempered martensite surface layer for both D20 and D32, respectively.

The average tempered martensite depth (Md), rim distance, is found to increase with increasing rebar diameter. In Table 3 the Md for the D20 mm rebar is ~1.52 mm whereas for the D32 mm rebar is ~1.9 mm. However, the tempered depth to the rebar diameter ratio is almost constant and ranging from 0.05–0.08, indicating that the water amount is proportional to rebar diameter. In addition, quenching time, as a rolling speed dependent, is generally known to affect the rim depth. Therefore, both water amount and time are modified for each rebar diameter while maintaining the rolling speed as in initial hot-rolling design. Table 3 also shows the effect of rebar diameter (*D*, mm) on the tempered martensite depth (*Md*, mm), tempered martensite volume fraction (*Mv*, %), and tempering temperature (*Te*, °C) values of different steels composition. *Mv* can be calculated as follows:

$$Mv, \% = [1 - 4\left(\frac{Rm, \text{ mm}}{D, \text{ mm}}\right)^2] * 100 \tag{6}$$

where, *Rm* and *D* are represented in Figure 4. For the B500B steel grade, changing the steel composition mainly increasing the Mn content besides V-addition and adjusting the process-parameters could be the reason behind the higher martensite depths observed in Table 3. The second phase precipitate of VC or V(CN) are expected to form during the austenite to ferrite transformation and these formations can enhance the ferrite grain refinement through increasing the potential nucleation sites. The addition of V slightly reduces ferrite grain size from 7 to ~4.5μm. However, in a rather small volume fraction, upper bainite appears in the microstructure of the core zone and it has been indicated that the breaking cementite lamellar into a small fragment is mainly due to a decrease in transformation temperature [30].

**Table 3.** The effect of different steel compositions on the formed tempered zone for rebars.

| Steel Grade | 20 mm (13 m/s: Rebar Rolling Speed) | | | | 32 mm (7 m/s: Rebar Rolling Speed) | | | |
|---|---|---|---|---|---|---|---|---|
| | Md, mm | Mv, % | Te, °C | Quenching Time, s | Md, mm | Mv, % | Te, °C | Quenching Time, s |
| B400B-R | 1.52 | 25.73 | 670 | 1.06 | 1.90 | 22.53 | 670 | 1.57 |
| B500B | 1.76 | 29.44 | 655 | 1.06 | 2.25 | 26.15 | 655 | 1.57 |

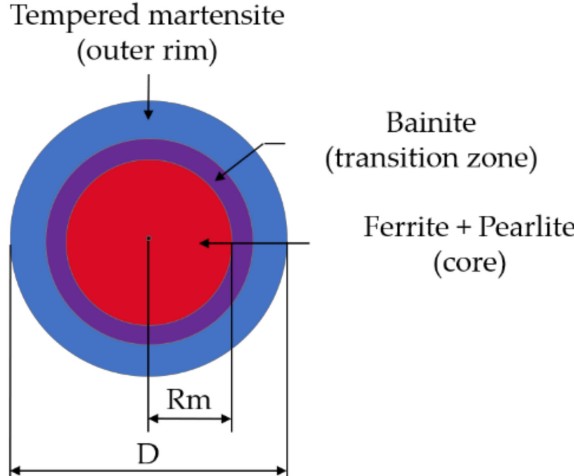

**Figure 4.** Schematic presentation of a cross-section of Tempcore rebar showing the different outer-surface rim and core zones with relative dimensions.

### 3.2.2. Core Zone

The microstructures of the core zone for both the steel grades B400B-R and V-alloyed B500B are shown in Figure 5. The optical micrographs generally revealed ferrite-pearlite cores for both diameters of each grade, however, the cores of D32 for both steels show relatively larger amounts of pearlite and coarser ferrite grains. Such coarser ferrite grains developed in the core of D32 rebars for both grades can be explained by the relatively higher core temperature (as indicated by the developed thermal profile using the thermal model) attained by applying low rolling speed. Processing of D32 rebars involves also slower cooling rate of the core renders the pearlitic phase transformation to proceed resulting in relatively larger amount of pearlite compared with that developed during processing of D20 rebars. More detailed microstructure characteristics of core zones for V-alloyed B500B of D20 and D30 rebars are represented in Figure 6, which reveals the typical lamella structure of pearlite. The representative areas depicted in Figure 6a,b (D20 and D32, respectively) indicate a larger pearlite colony of D32 that developed during processing of D20.

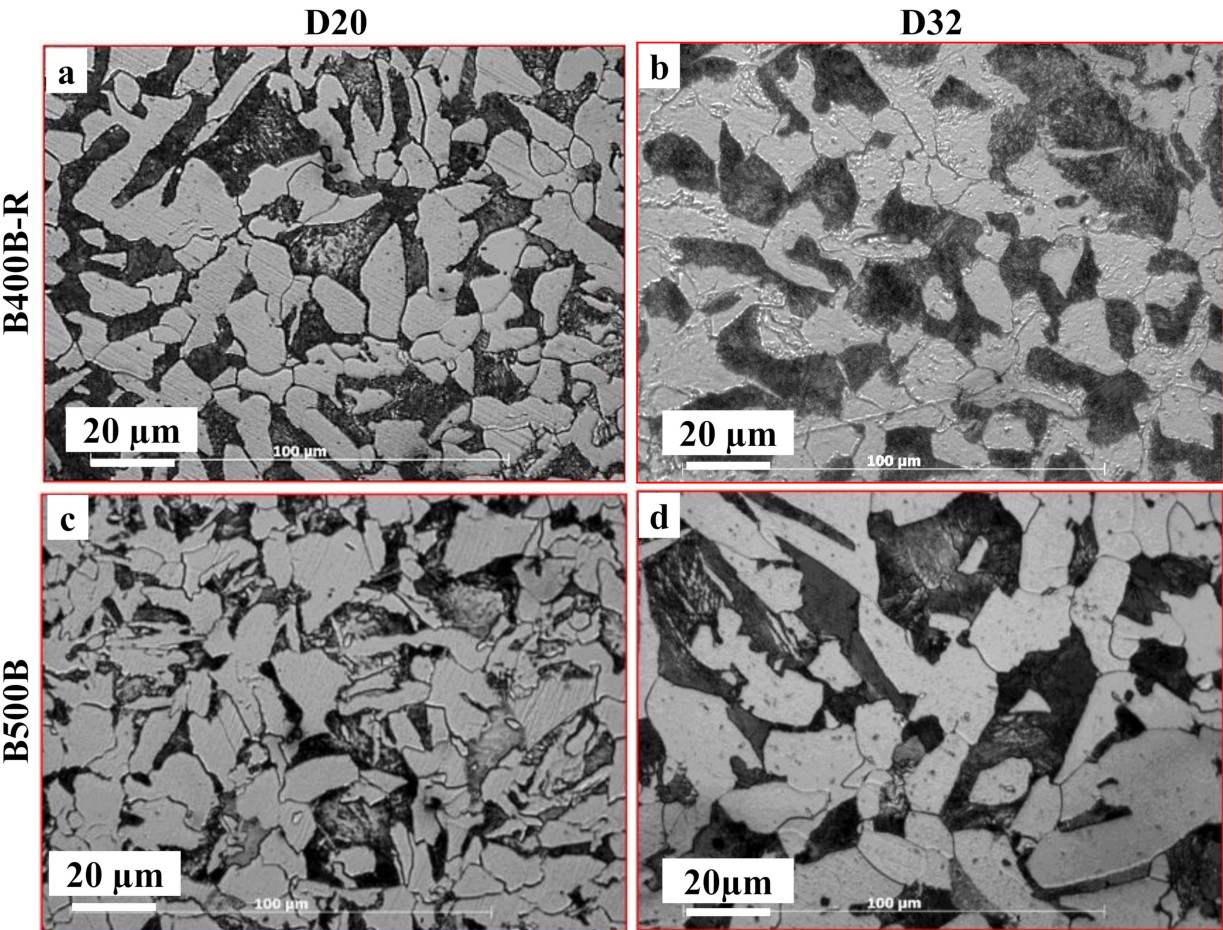

**Figure 5.** Optical micrographs showing the developed microstructures in the core zones (**a**,**b**) for D20 and D32 of B400B-R grade. (**c**,**d**) show the microstructure in the core zones for D20 and D32 of B500B steel grade.

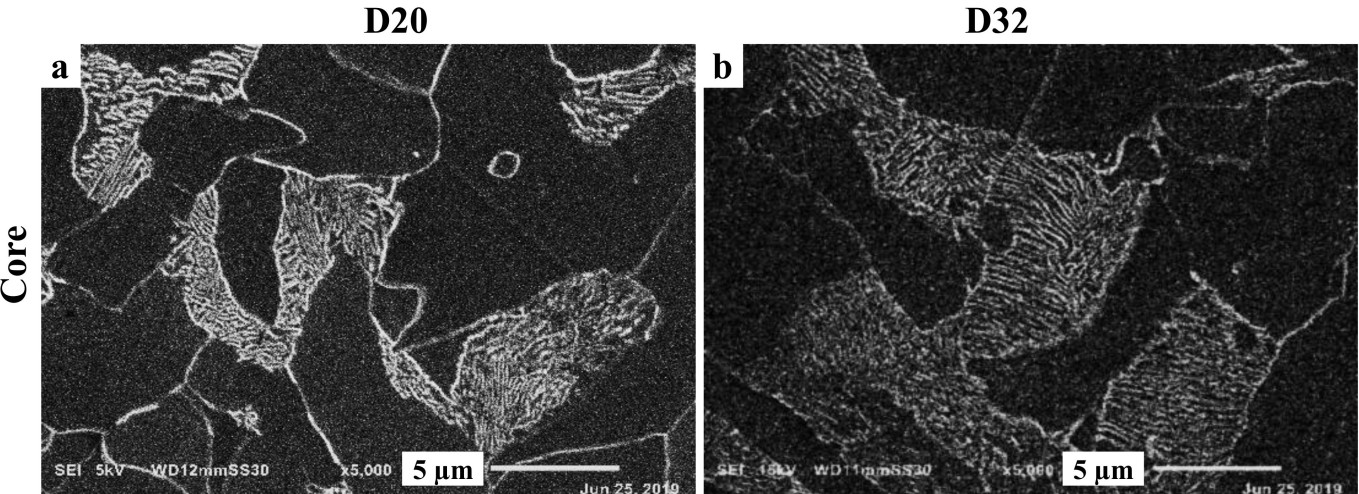

**Figure 6.** SEM micrographs of B500B steel grade. (**a**,**b**) indicate ferritic-pearlitic microstructures developed in the core zones for both D20 and D32, respectively.

*3.3. Mechanical Properties*

3.3.1. Hardness Profile

The hardness measurements from the rim surface to the core for the studied steel rebars are shown in Figure 7. A drop in the hardness values from surface to the rebar center is generally observed. This result is expected and mainly attributed to the high cooling rate, which gradually decreases toward the rebar center. The formation of self-tempered martensite increases the hardness of the surface layer that is found to vary with increasing rebars diameters, a value of 222 decreases to 198 HV with increasing B400B-R steel diameter from D20 to D32 mm, respectively, and a value of 275 decreases to 246 HV with increasing B500B steel diameter from D20 to D32 mm. Similar trend is also observed for the minimum values of the core hardness, as shown in Figure 7. The measured low hardness values can be associated with the mixed microstructure of ferrite and pearlite formed in the core. The addition of Vanadium to steel produces a less steep drop in hardness values toward the rebars core. The change of the hardness values in the transition zones, between tempered surface and core, occurred with high rate, being rebars diameter dependent, and hence formation is a cooling rate dependent. These variations in hardness values can be attributed to different volume fractions of the consistent phases in the transition zone. As mentioned above these transition zones have been identified to consist of bainite, AF, ferrite, and cementite and all contributes to the measured hardness values. These zones in addition to the tempered martensite are expected to increase the strength level of the manufactured rebars [31]. The fine microstructure of both areas is a further benefit for the steel toughness, since it provides a high resistance to crack propagation [27,32], and furthermore the existing mixture in the soft core can also be advantageous for improving the impact toughness of rebars.

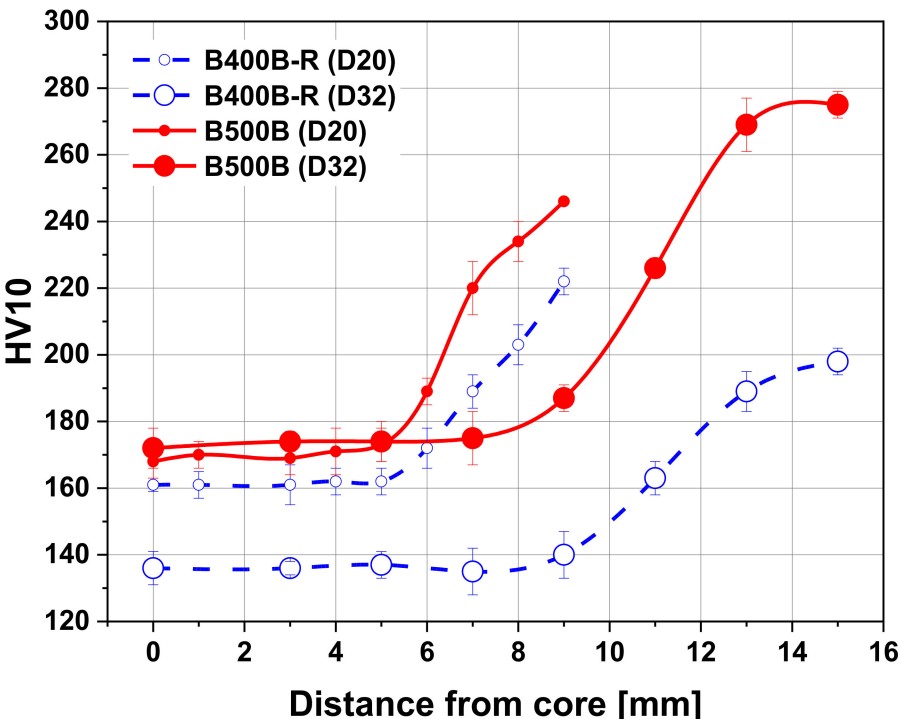

**Figure 7.** Vickers hardness distribution from the center to the surface of B400B-R and B500B steel grades.

### 3.3.2. Tensile Strength

The strength of steel rebars, rapidly quenched, depends on the volume fraction of each microstructure region in the cross section and they are separately contributing to the overall measured strength. Higher strength contribution can be attributed to the higher area fraction of tempered martensite rim. The measured yield strength, ultimate strength and total elongation values of the Tempcore rebars steels are shown in Figure 8. The obtained results agree with the international and national standards requirements. The estimation of yield strength for QTB rebars is related to the measured rim thickness and its hardness through an empirical equation [33]:

$$YS = \frac{[k1 * rim\ hardness + k2 * core\ hardness + k3 * average\ rim\ thickness]}{3} \qquad (7)$$

where $YS$ is the yield strength (MPa), rim and core hardness in MPa, average rim thickness in mm, and $k1$, $k2$ and $k3$ are constants. The relation is limited to low carbon steel with carbon content up to 0.30%, 0.05–1.2% Mn and <0.30% alloying elements. The constants $k1$, $k2$, $k3$ are taken as 2.130, 2.350, 203.034 respectively [33].

Compared to measured values, the calculated yield strength is slightly lower. For example, for the D20 mm B400B-R rebar with 1.52 mm rim thickness, the yield strength calculated using Equation (7) is ~401 MPa against ~445 MPa experimentally measured, while for the D32 mm B400B-R rebar with 1.9 mm rim thickness, the yield strength calculated using Equation (7) is ~388 MPa against ~413 MPa experimentally measured, see Figure 8. The difference can be attributed to the absence of the transition zone effect in Equation (7). Microstructure features such as grain size, precipitations and dislocation density are essential for reaching the required mechanical properties. The impact of microstructure effect on rebar mechanical properties can be evaluated using the corresponding ultimate tensile strength (*UTS*) to the yield strength *YS* ratio [*UTS/YS* = *UYR*]. This ratio is currently used to define the type of steel application within certain corresponding limits, e.g., the ASTM A706,2006 standard, in this respect limit of 1.25 ratio, is selected for the seismic rebar resistance. The different steel composition rebars indicate that micro alloying vanadium

is very effective and decreases the *UYR* ratio from around ~1.3 to ~1.22 in D20 mm, and ~1.35 to ~1.17 in D32 mm rebar when vanadium is added (~0.067%) to steel B500B. Such a decrease in the *UTS/YS* ratios could also be attributed to the range of carbon content with the vanadium in the steel composition. A low *UYR* value can be an indication that there is a high resistance to impact fracture under load [34]. This behavior is related to the presence of some microstructure features such as bainite, ferrite and/or martensite, leading to a higher resistance to dislocation movements and reducing the material uniform elongation. However, it has been reported that the formation of a small fraction of a ductile phase in the hard phases can largely improve the yield strength and *UYR* ratio without altering the uniform elongation [35]. The vanadium content in the steel could contribute to ferrite grain refinement through retarding recrystallization of austenite during the hot deformation stage. Moreover, during hot deformation the accumulated defect can enhance the intergranular precipitates of VC or V(C,N) formation [36]. It was reported that the addition of 1 wt.% vanadium to X20CrNiMnVN18-5-10 steel resulted in increasing the yield strength to ~600 MPa through an interplay among several strengthening mechanisms, namely, solid solution, Hall-Petch effect and Ashby-Orwan effect [37]. Moreover, the type and size of V(C,N) precipitates can control not only the mechanical properties and deformation mechanism [38] but also the corrosion and hydrogen embrittlement behaviors [39].

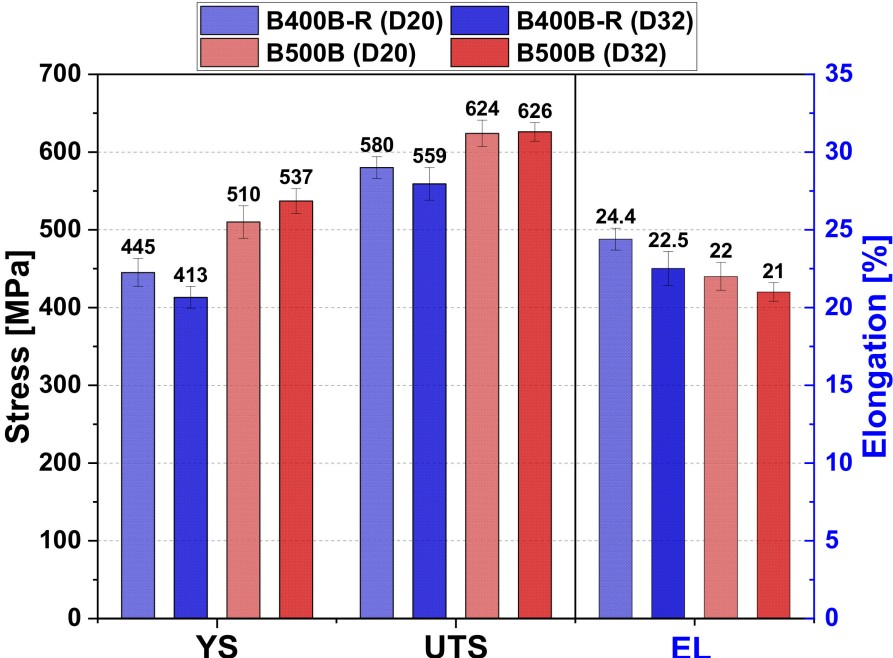

**Figure 8.** Evaluated tensile properties of B400B-R and B500B steel grades represented in bar chart for D20 and D32 rebars. *YS*, *UTS* and El stand for yield strength, ultimate tensile strength and percent of total elongation, respectively.

### 3.4. Evolution of Thermal Cycles

The thermal profiles of studied rebars, starting from the cooling box entry up to the equalization temperature, are shown in Figure 9. This is performed by discretizing the radius of the rebar's cross section into segments of equal dimensions. A mid node for each segment is selected at which the temperature is captured and recorded. The thermal profiles show the temperatures of the rebars' cross sections at the surface down through the core. Only couple of internal nodes' temperature profiles are plotted for clarity. The predicted lowest temperature reached by quenching is found to vary from 298 to 148 °C with increasing the rebar diameters from 20 to 32 mm, respectively, Figure 10. The temperature reached for the 32 mm rebar is beyond the martensite-finish temperature (Mf) as a result, a complete transformation can be expected in the rim zone. On the other hand, only about 50% of martensite transformation can be achieved for the 20 mm rebar diameter.

Figure 10 also shows the increase of the generated temperature values from the minimum quenching temperature up to the equalization temperature (Te). This rise in temperature is generally related to the heat flow effect from the core to the surface causing the tempering of the martensite zone. A close agreement between calculated Te and the measured one for the different rebar diameters is shown in Figure 11.

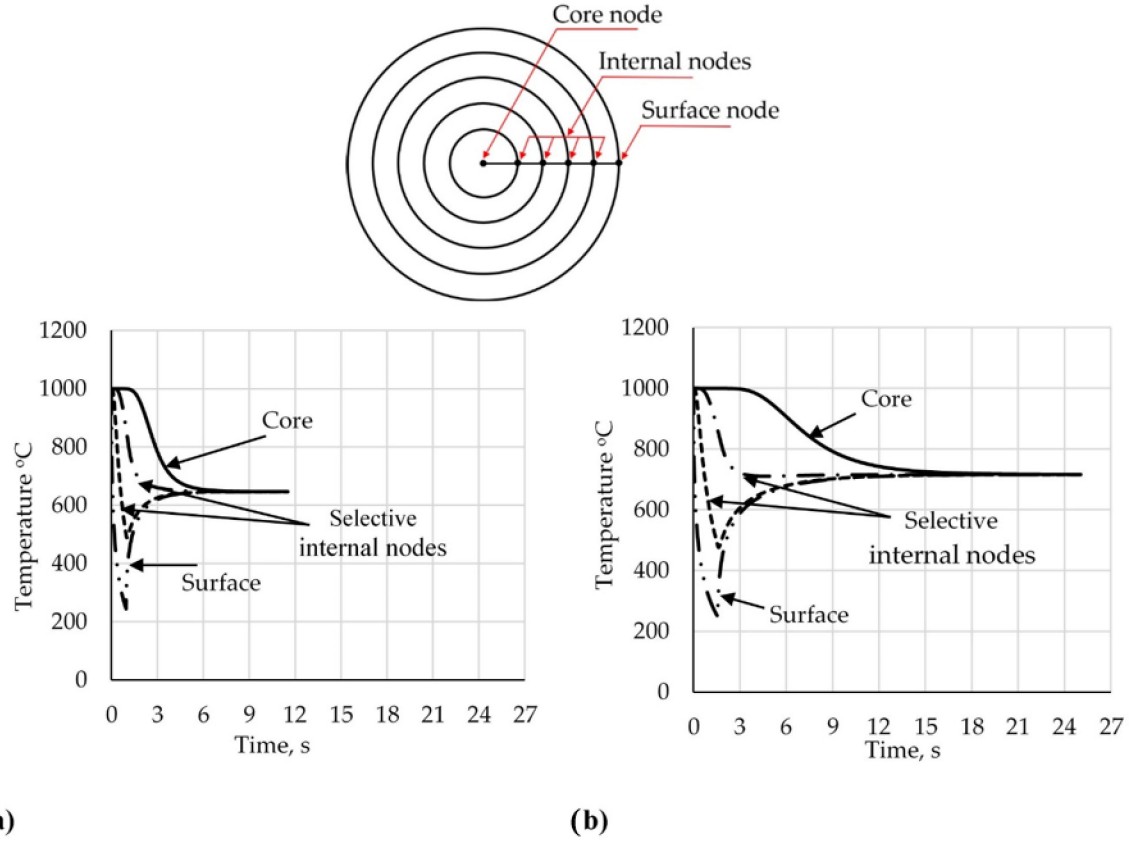

**Figure 9.** Evolution of thermal cycles in the core as well as surface zones during processing of D20 (**a**) and D32 (**b**) rebars.

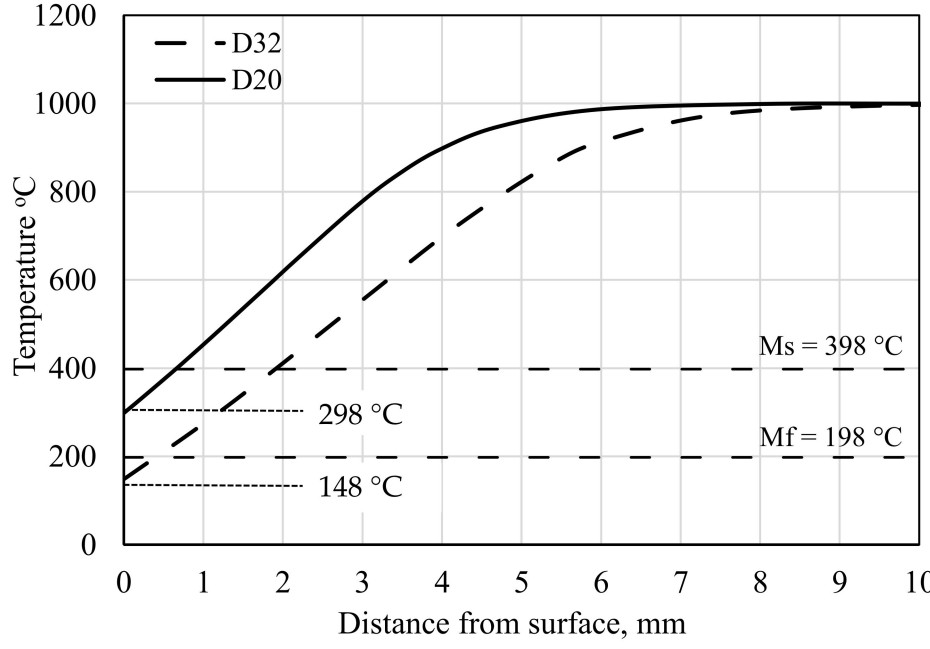

**Figure 10.** Temperature distribution after quenching for the studied rebars.

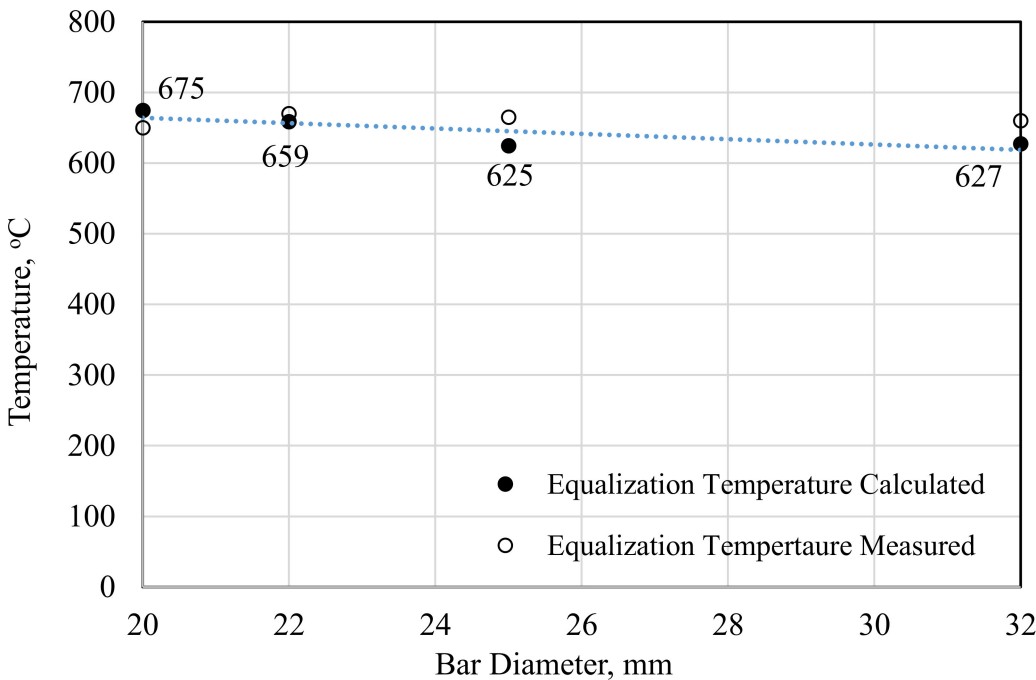

**Figure 11.** Equalization temperature after self-tempering process: 4 diameters, 20, 22, 25, and 32 mm rebars are industrially produced, but onle two rebars (20, and 32 mm) are represented in this article.

The extended time to reach Te from the minimum quenching temperature also increases with an increasing rebar diameter. It is likely that such a time will not affect only the martensite zone but also the transition zone where tempered martensite has been observed for the first layer. The former microstructure analyses, in this work, show a multi-phase structure formation in the transition zone along with their volume fraction, which is a rebar diameter dependent, i.e., depends on the rate of cooling.

*3.5. Precipitation Kinetics*

The precipitation kinetics of V(C,N) for B500B grade was simulated based on the thermal profiles predicted using the thermal model for different diameters i.e., D20 and D32 rebars. The thermal profiles for the outer surfaces as well as for the cores of the corresponding diameters (Figure 9) are set as the heat-treatment cycles for precipitations kinetics simulations. Figure 12 shows the resulting fraction of precipitates vs. time curves for the predicted thermal cycles, namely, outer surface D20, outer surface D32, core D20 and core D32. During the in-line cooling practices in the QTB the equalization temperature is the main controlling parameter that can define the extent of precipitation. The kinetics simulation indicates a variation in precipitation potential depending on the processed diameters. Obviously, the D32 rebars manifest a greater potential to form higher volume fractions of V(C,N) precipitates in both of outer surface and core as well. The fractions of V(C,N) precipitates reach approximately 0.0007, 0.001, 0.0008 and 0.0014 for outer surface D20, outer surface D32, core D20 and core D32, respectively. Such higher potential for precipitation during processing of D32 is emerging from: (1) the fact that the eventually attainable equalization temperature during processing of D32 rebars renders higher surface and core temperatures, 655 and 750 °C respectively, than those could be achieved during processing of D20 rebars (eventually 620 °C); (2) the precipitation kinetics generally shows a sluggish behavior as the temperature drops implying a larger time-window available for precipitation when the equalization temperature is high. It is worth mentioning that the fraction of precipitation approaches the theoretical equilibrium limit (0.0015 as indicated from the thermodynamic calculations) during processing of D32 rebars, however, the precipitation potential is not fully exploited during the processing of D20 under the current applied operational parameters.

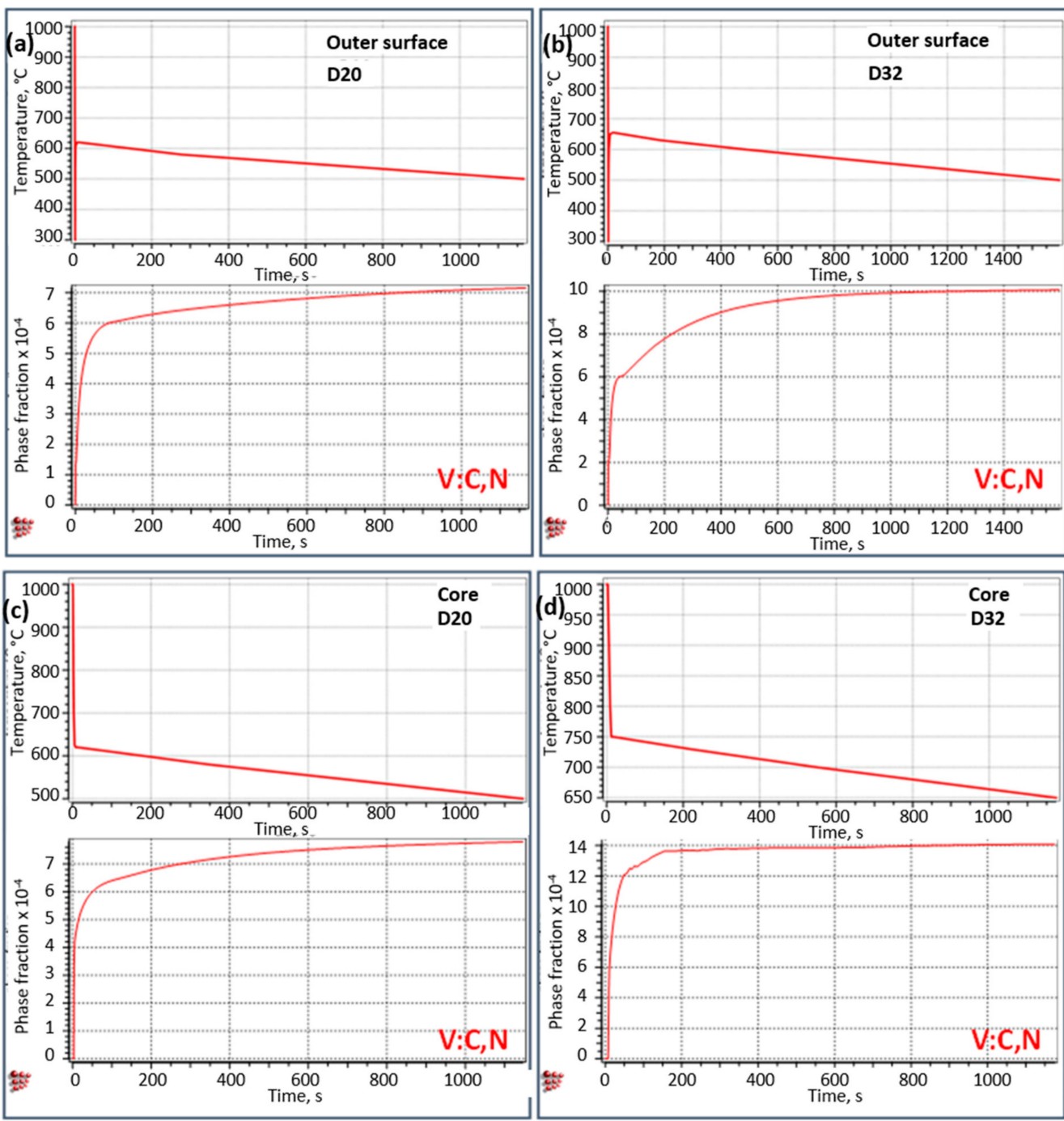

**Figure 12.** The precipitation kinetics during processing of D20 and D32 rebars for V-alloyed B500B steel grade. The thermal cycles and fraction of precipitation vs. time curves are represented in (**a**–**d**) for the outer surface D20, outer surface D32, core D20, and core D32, respectively.

## 4. Conclusions

An optimized set of in-line processing parameters during Tempcore practices including hot-rolling speed of 13 and 7 m/s, water flow rate of 600 and 620 m³/h and equalization temperature of 660–670 °C were utilized to industrially manufacture two different rebars D20 and D32, respectively, for two steel grades, namely, B400B-R and 0.067 wt.% V-alloyed B500B. V-alloyed B500B grade aims to improve its whole mechanical property profile through V(C,N) precipitation during adopting processing parameters that render higher productivity than that can be obtained under normal conditions in the

quenching and tempering box (QTB). The following conclusions can be drawn based on the current findings:

—  Typical graded microstructures consisting of soft ferrite-pearlite core encapsulated with hard tempered martensite surface were observed for both diameters of each steel grade.

—  For both of B400B-R and B500B grades, the adopted speed of hot rolling-line of 7 m/s to manufacture D32 rebars is responsible for increasing the quenching time in the QTB and the corresponding increase in the tempered martensite depth compared with its counterpart during manufacturing D20 rebars processed by 13 m/s as the speed of rolling-line.

—  For B400B-R grade, D32 rebars show relatively softer microstructure with lower yield and tensile strength values than D20 rebars exhibit, which can be explained by the higher attainable core temperature for D32 as indicated from the thermal model. Such high core temperature leads to a larger degree of tempering for the martensite surface layer and relatively coarser ferrite-pearlite core, which were reflected by lower hardness values.

—  For V-alloyed B500B grade, an opposite behavior than that of B400B-R grade is observed. The higher attainable core temperature in case of processing D32 rebars renders the exploitation of full potential of V(N,C) precipitation possible in both of the core as well as the surface zones, which allows increasing the hardness, yield strength and tensile strength values compared with their counterparts in case of D20 rebars that experience relatively short precipitation time at relatively lower temperature.

—  The V-addition to B500B steel grade does not scarify the attainable total elongation in comparison to the B400B-R steel grade, since the observed differences are below 2.5%.

—  The current work emphasizes the additional advantages of tuning the microstructure and mechanical properties that can be realized from V-alloyed rebars processed by Tempcore practices, when the suitable equalizing temperature and quenching time can be properly adjusted to control and exploit the full potential of precipitation process. In the future work, the electrochemical behavior and bending characteristics of such rebars will be carefully investigated.

**Author Contributions:** Conceptualization, E.A., S.I. and T.A.; methodology, E.A., S.I.; software, S.A.E. and T.A.; formal analysis, E.A., S.I., S.A.E. and T.A.; investigation, M.G.; resources, E.A. and T.A.; writing—original draft preparation, E.A. and S.I.; writing—review and editing, E.A. and T.A.; visualization, E.A., S.A.E. and T.A.; supervision, S.I. and T.A.; All authors have read and agreed to the published version of the manuscript.

**Funding:** This research received no external funding.

**Data Availability Statement:** The data used to support the findings of this study are available from the corresponding authors upon request.

**Conflicts of Interest:** The authors declare no conflict of interest.

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
