# Peer review of "Microstructure and Mechanical Properties of V-Alloyed Rebars Subjected to Tempcore Process"

_metals, doi:10.3390/met11020246_

Round 1
Reviewer 1 Report
This manuscript compares the microstructures and mechanical properties of two rebars B400B-R and B550B with two different diameters of 20mm (D20) and 30mm (D32). A thermal model was developed to describe the cooling and heating process. Compared with the B400-R steel, the V-alloyed B500B shows different strength characteristic between D20 and D32. V(C,N) precipitations are believed to be the reason steel for the strength difference with the help of the thermal model. Despite all these efforts, this manuscript is more like a technical report than a scientific paper owing to the following reasons:
- No evidence of direct observation for the V(C,N) precipitations is provided, which weakens the author’s explanation for the strength increase. Though precipitation fraction was calculated at different conditions, it has not been verified by experimental results. Without the accurate information of the precipitation size and fraction, it is difficult to judge to what extent the V(C, N) particles affecting the strength increase.
- Several processing parameters are mentioned, but their inter-relationship is not clearly explained in the context, such as cooling time, quenching time, quenching intensity, etc., leading to inevitable confusion and difficulty in understanding.
- Many spelling and grammar mistakes appear in the manuscript, such as in line 148, 162, 191, 237, 247, 402, and so on.
- The scales in Figure 2 are not the same format, and they are not in a proper length. The scales in Figure 3 are not in the same length.
- Figure 5 mentioned in line 245 and 246 should be Figure 6 according to context. Much more care should be paid in preparing the manuscript.
- The caption of Figure 5 is difficult to understand.
- Some results lack the error range, for example, figure 7 and figure8.
Consequently, the reviewer does not recommend this manuscript for publication.
Author Response
Ms. Rena Wang
Assistant Editor, MDPI
Manuscript ID: metals-1075538
Title: “Tailoring the microstructure and mechanical properties of V-alloyed
rebars during in-line tempcore processing”
Dear Ms. Rena Wang
On behalf of all the contributing authors, I would like to express our sincere appreciations of your letter and reviewers’ constructive comments concerning our article entitled “Tailoring the microstructure and mechanical properties of V-alloyed
rebars during in-line tempcore processing” (Manuscript ID: metals-1075538). These comments are all valuable and helpful for improving our article. According to the reviewers’ comments, we have made extensive modifications to our manuscript and supplemented extra data to make our results convincing. In this revised version, changes to our manuscript were all highlighted within the document by using track changes function. Point-by-point responses to the reviewers are listed below this letter.
Reviewer#1
This manuscript compares the microstructures and mechanical properties of two rebars B400B-R and B550B with two different diameters of 20mm (D20) and 30mm (D32). A thermal model was developed to describe the cooling and heating process. Compared with the B400-R steel, the V-alloyed B500B shows different strength characteristic between D20 and D32. V(C,N) precipitations are believed to be the reason steel for the strength difference with the help of the thermal model. Despite all these efforts, this manuscript is more like a technical report than a scientific paper owing to the following reasons:
Response: The authors would like to thank the reviewer for the constructive criticism and the valuable comments, which allow them to further improve the quality of the manuscript and to sharpen its scientific discussion. Also, they would like to emphasize the scientific background of this manuscript. There were several scientific and applied-research questions behind this work, for example, this one related to the current manuscript “why does V-addition cause an improvement in mechanical properties of large rebar sizes but not of the small ones? The answer of this question seems to be intuitive at the lab-scale experiments, where we control each individual process parameter (e.g. deformation, temperature, time, cooling, etc.). However, it is not such easy to be answered when there will be a complex interplay due to several changes in process parameters at once during industrial hot-rolling processes of rebars. We believe that the key answer of this question is the control of precipitation events during processing, therefore, we transferred the industrial process parameters to a research-based thermal model that considers the treat the complex interplay of the process parameters in the quenching and tempering box (QTB) during Tempcore process of various rebar sizes. The resulting thermal profiles were used to follow the precipitation kinetics in different proposed scenarios in QTB. Combining the results of kinetics precipitation with microstructure evolution has led us to interpret the observed opposite mechanical behavior of V-alloyed rebars resulting from changing process parameters to manufacture different diameters. Furthermore, the current manuscript provides a scientific-based recipe to how to exploit the full potential of V-addition in order to improve the performance of different size rebars during such a widely used Tempcore technology. Depending on this short rebuttal, we believe that the current manuscript not only deliver an industrial and technical insight, but also adds a value for the scientific community in field of rebar manufacturing.
- No evidence of direct observation for the V(C,N) precipitations is provided, which weakens the author’s explanation for the strength increase. Though precipitation fraction was calculated at different conditions, it has not been verified by experimental results. Without the accurate information of the precipitation size and fraction, it is difficult to judge to what extent the V(C, N) particles affecting the strength increase.
Response: The authors completely agree with the reviewer’s opinion that without quantification of precipitations it would be hard to judge their influence on the increase of strength. Since the mechanical properties of all rebar sizes achieve the minimum required standard strength levels, the authors brought the focus on the exploitation of full-potential of V-addition to further improve the mechanical performance not to quantify the effect of V-based precipitates. This idea is very clear depending on the thermal model and the kinetics of precipitations. Nevertheless, the authors are fully convinced with the reviewer’s recommendations, and accordingly they integrated the conclusions of their own previous studies on other V-alloyed advanced high strength steels of second and third generations (like V-alloyed medium manganese steels) to elucidate the influence of precipitates characteristics on not only mechanical but also on other environmental properties like corrosion and hydrogen embrittlement behaviors. The following text was integrated in the discussion: “It was reported that the addition of 1 wt.% vanadium to X20CrNiMnVN18-5-10 steel resulted in increasing the yield strength to ~600 MPa through an interplay among several strengthening mechanisms, namely, solid-solution, Hall-Petch effect and Ashby-Orwan effect [37]. Moreover, the type and size of V(C,N) precipitates can control not only the mechanical properties and deformation mechanism [38] but also the corrosion and hydrogen embrittlement behaviors [39].”
- Several processing parameters are mentioned, but their inter-relationship is not clearly explained in the context, such as cooling time, quenching time, quenching intensity, etc., leading to inevitable confusion and difficulty in understanding.
Response: The authors thank the reviewer for this carful comment that improves the readability of the manuscript. In fact, we used all the mentioned terms for the same meaning, however, these terms were unified throughout the manuscript to eliminate the confusion.
- Many spelling and grammar mistakes appear in the manuscript, such as in line 148, 162, 191, 237, 247, 402, and so on.
Response: The authors appreciate the reviewer’s comment, and did an extensive language-correction through the whole manuscript.
- The scales in Figure 2 are not the same format, and they are not in a proper length. The scales in Figure 3 are not in the same length
Response: The Figure was modified according to the reviewer’s remark.
- Figure 5 mentioned in line 245 and 246 should be Figure 6 according to context. Much more care should be paid in preparing the manuscript.
Response: The authors do apologize for this mistake, and thanks for the carful observation of the reviewer.
- The caption of Figure 5 is difficult to understand.
Response: The authors appreciate the reviewer’s remark and accordingly, did modification on the caption of Figure 5
- Some results lack the error range, for example, figure 7 and figure8.
Response: Figure 7 and 8 are modified to meet the reviewer’s requirements.
Reviewer 2 Report
Dear Authors,
Tempcore technology is well known and widely used all over the world but some modification to improve performance characteristics of the rebars at acceptable cost level are still the subject of investigation.
All my suggestions and questions to your paper I marked in the attached file.
Thank you
Best regards

Author Response
Ms. Rena Wang
Assistant Editor, MDPI
Manuscript ID: metals-1075538
Title: “Tailoring the microstructure and mechanical properties of V-alloyed
rebars during in-line tempcore processing”
Dear Ms. Rena Wang
On behalf of all the contributing authors, I would like to express our sincere appreciations of your letter and reviewers’ constructive comments concerning our article entitled “Tailoring the microstructure and mechanical properties of V-alloyed
rebars during in-line tempcore processing” (Manuscript ID: metals-1075538). These comments are all valuable and helpful for improving our article. According to the reviewers’ comments, we have made extensive modifications to our manuscript and supplemented extra data to make our results convincing. In this revised version, changes to our manuscript were all highlighted within the document by using track changes function. Point-by-point responses to the reviewers are listed below this letter.
Reviewer#2
The authors really appreciate the carful comments and thorough corrections made to the manuscript by the reviewer. We followed all recommendations and did accordingly major changes as follow:
- Title: it was changed as recommended. It reads in the new version “Microstructure and Mechanical Properties of V-alloyed Rebars Subjected to Tempcore Process”
- Abstract and Keywords: The authors followed the suggested corrections to and adjust them accordingly.
- List of Nomenclature and abbreviations: the list was deleted and as recommended, since they are all repeated in the test.
- Introduction: the authors appreciate the reviewer’s comments and did the suggested changes, which are:
- Martensite start (Ms) temperature.
- A short explanation of the transition zone (containing upper or lower bainite) was inserted as recommended. “Between the surface and core zones a transition zone develops, where the cooling rate could be sufficient to trigger the lower or upper bainitic phase transformation depending on the applied process parameters.”
- The carbon-dependency of martensite’s strength and ductility was adjusted according to the reviewer’s correction. “Tempering results in stress relief due to the diffusion of carbon out of the martensite depending on the carbon content”.
- Several typo corrections were made as revised by the reviewer.
- Materials and Methods:
- The authors would like to thank the reviewer for the useful suggestions to table 1 and table 2. We rearranged the sequence of N and Ceq. Also, we inserted the Ceq-equation with the corresponding reference.
- The inaccurate terminologies such as mechanical characteristic values and elongation were corrected (mechanical properties and total elongation) according to the reviewer recommendation.
- The authors thank the reviewer for the important question regarding whether there are changes in the temperature of the surrounding medium (during long-time process. We understand the reviewer’s concern and we expect also minor changes of this temperature over time during processing, however, the cooling system in the QTB is designed in such a way that the temperature of surrounding medium is maintained almost the same by means of an integrated recycling and feeding water system. Accordingly, we assumed a constant value of . This assumption also reduces the complexity during building up the thermal model.
- The authors would like to thank the reviewer for the valuable question regarding whether there were changes in water temperature during long-process. It is true that there will be a temperature profile along the billet (bar) length after its exit from the casting strand, which should be minimized during reheating in reheating furnace before hot-rolling guiding the billet (bar) into the hot-rolling mill. However, the authors mistakenly used the term bar (that understood as billet) instead of rebar. Accordingly, we corrected this mistake throughout the manuscript to avoid inconsistencies and discrepancies in meaning.
- Results and discussion:
- The sequence of insets of Figure 1a and b was changed as recommended by the reviewer.
- The authors thank the reviewer for the meticulous comment regarding microstructure observation. we admit that the applied magnification using LOM and even SEM was not enough to resolve the precipitates that their observation requires applying higher resolution technique such TEM, which was and still not accessible to the authors. However, major changes were made to the text to adjust it with both of the LOM and SEM observations represented in Figure 2 and 3. The following changes were made to the text: “From microstructural examination, a tempered martensite with lath like morphology is observed. Figure 2 shows that the grain structure of martensite is consisting of differently oriented martensite packets containing various blocks of martensite lathes. Such typical martensitic microstructure with three level-hierarchy i.e., packets, blocks and lathes were reported by Krauss [28]. The pronounced variation in orientations of the developed martensitic microstructure emerges from the several possible crystallographic orientation relationships between martensite and the parent phase austenite. Although the Kurdjumov-Sachs (K-S) orientation relationship model considers 24 variants to form martensite from austenite grains and 6 possible variants in each packet. Kitahara et al. [29] pointed out that neither all the 24 austenite grain variants nor the 6 packet variants can essentially appear. A narrower lath is obtained for the smaller 20 mm rebar as shown in Figure 2(a, c) attributable to the relatively lower equalization temperature achieved after during process. The SEM micrographs of the outer layer of the B500B steel grade shown in Figure 3 (a, b) indicate the possible carbonitride precipitates during Tempcore process. The boundaries of martensite packets seem to be decorated with fine carbonitride precipitates, which could not be resolved at the applied magnification.”
- The authors thank the reviewer for the carful observation, and accordingly, the font size of the length scale of Figure 2 is now visible.
- We agree with the reviewer that the increase in hardenability is not only due to V-addition. We deleted this statement and modified the text accordingly. „For the B500B steel grade, changing the steel composition mainly increasing the Mn content besides V-addition and adjusting the process parameters could be the reason behind the higher martensite depths observed in Table 3“
- The authors thank the reviewer for the comment on the schematic representation in Figure 4. A revised schematic representation of the developed different zones is available now considering the reviewer’s recommendations.
- The authors really appreciate the reviewer’s useful typo and style revisions. The several detected typo and style errors were corrected as recommended.
- The scale bars for the micrographs of Figure 5 are now visible. The authors thank the reviewer for detecting the deficiencies in the representations. Regarding the possibility of doing TEM, the author admit its significance in supporting the research idea through visualizing the V(N,C) precipitates to interpret their influence on the evaluated mechanical properties, however, the authors had and still have no access to such high resolution technique. Therefore, the authors are asking for understanding of the reviewer.
- The reviewer’s question about the analysis of uniform elongation values for the applied production process is a valuable one, however, the authors represented the standard mechanical properties according to the specified analysis according to the corresponding ASTM A706 standard.
- The typo in Figure 8 was corrected. Thanks for the reviewer.
- We followed the reviewer’s recommendations and revised Figure 9, 10 and 11 and did the suggested modifications.
- Conclusions:
- The authors followed the reviewer’s suggestions to improve the readability of the drawn conclusions. Moreover, two conclusions addressing the total elongation and the increasing of costs due to V-addition were added as the reviewer advised.

Reviewer 3 Report
See attachment "Comments and Suggestions for Authors
"

Author Response
Ms. Rena Wang
Assistant Editor, MDPI
Manuscript ID: metals-1075538
Title: “Tailoring the microstructure and mechanical properties of V-alloyed
rebars during in-line tempcore processing”
Dear Ms. Rena Wang
On behalf of all the contributing authors, I would like to express our sincere appreciations of your letter and reviewers’ constructive comments concerning our article entitled “Tailoring the microstructure and mechanical properties of V-alloyed
rebars during in-line tempcore processing” (Manuscript ID: metals-1075538). These comments are all valuable and helpful for improving our article. According to the reviewers’ comments, we have made extensive modifications to our manuscript and supplemented extra data to make our results convincing. In this revised version, changes to our manuscript were all highlighted within the document by using track changes function. Point-by-point responses to the reviewers are listed below this letter.
Reviewer#3
The authors would like to thank the reviewer for his valuable and constructive comments. The responses for these comments are listed as follows:
-Line 3, line 10 (tempcore), line 38 (Tempcore) and so on: Tempcore is not equal in the article (with capitol or without capitol).
It was modified in overall the manuscript as “Tempcore”
-Line 14, line 108, line 236 (“optical microscope”) “light microscope”
They were modified in overall the manuscript as “light optical microscope”
-Line 18 "outer martensite" is not correct, it should be "outer tempered martensite" or "selftempered martensite", lines 20, 22, 26 also and so on in the article.
They were modified in overall the manuscript as “outer tempered martensite”
-Line 17 V(N,C), line 27 V(C,N), line 31 V(N,C), List of Nomenclature and Abbreviationin M(C,N), line 81 V(N,C) and so on in the article.
They were all unified as V(C,N).
-Carbide phases were not analyzed in the article. In my opinion it would be better in the article write "carbide or carbonitride M(C,N) phase on the vanadium base" because in the carbide or carbonitride phase is not only one metallic element vanadium. There will be small quantity of iron and probably chromium.
The authors really appreciate the reviewer’s suggestion, and they also agree that many carbides contain iron and chromium in their composition, however, the thermodynamic calculations in this study confirmed the formation of V(C,N) type of precipitates with relatively low amount of only iron matrix as chromium is not one of the alloying elements of the current investigated V-alloyed steel. The chemical composition of the V(C,N) over the process temperature-window is appended here for the reviewer’s kind consideration.
-List of Nomenclature and Abbreviation
Ms and Mf - "Martensit start" and "Martensit finish" would be better
(V(CN) Vanadium Carbide Nitride) V(C,N) vanadium carbonitride
This section “List of Nomenclature and Abbreviation” was deleted according to the second reviewer comment, and all abbreviations were explained in the text.
-Line 54 (…diffusion of carbon out of the extremely brittle but strong martensite.) would be better “…diffusion of carbon out of the martensite.”
It was done.
-Line 62 (“hard martensite”) “hard selftempered martensite”
It was done.
-Line 162 (oC) °C
It was corrected.
-Line 192 (…contain carbide precipitates, and which could be of the É›-carbide types.) could be more accurate “...contain precipitates.” or “contain carbonitride precipitaces”) or …
It was done.
-Line 203 (steel gradessteel grade) steel grade
It was deleted.
-Line 242 - grammatical mistakes
They were corrected.
-Line 245 (“Figure 5”) “Figure 6”
Line 246 (“Figure 5a and b”) “Figure 6a and b”
They were corrected.
-Line 250 Figure 5 Line segment 100 μm is too long. It would be better 10 μm segment.
It was modified to 20 µm segment, since 10µm appears to be too small.
-Line 268, line 319 (Vanadium) vanadium
It was modified.
Line 281 (Vicker) Vickers
It was modified.
-Line 297 (“…slightly higher”.) “…slightly lower.”
It was corrected.
-Line 298 (eleongation) elongation
It was corrected.
-Line 335 (martensite finished) martensite finish
It was corrected.
-Line 337 (“diamete”) “diameter”
It was corrected.
-Line 349 Figure 9 – several mistakes
They were corrected.
Line 365 In Figure 10 ( Ms=397.88) MS = 398 °C and so on.
They were modified.
-Line 394 In Figure 12 (V:C,N) V(C,N) or M(C,N)
They were modified.
-Line 411, 416 (martensite) “tempered martensite” or “selftempered martensite”
It was modified.
-Lines 400, 415, 416, 419, 420, 421, 423, 427 – typo
All were corrected.
We really appreciate the referee’s valuable comments to improve our manuscript.
Yours sincerely,
Essam Ahmed, on behalf of all co-authors.

Reviewer 4 Report
The paper investigates the effect of V addition and different diameter on mechanical properties and microstructure of Tempcore processed rebars.
I suggest the authors to modify the paper according to the following comments:
- Line 97: Word reference error. Please Fix it
- Line 102: Word reference error. Please Fix it
- SEM micrographs of figure 3 and figure 5 are lacking resolution. Please improve the resolution and add higher magnification micrographs showing V(C,N).
- Line 226: Word reference error. Please Fix it
- Line 229: typos error “…could be the reason behind the the higher martensite depths observed…”
- Line 230: Word reference error. Please Fix it
- Line 233: how did you measure grain size? Explain in materials and methods section
- Table 3: martensite volume fraction is quite accurate, how did you measure it? Explain in Materials and Methods section
- Line 252: typos error “…rendes the pealitic phase transformation to proceed…”
Author Response
The authors want to sincerely thank the Reviewer for his/her comments about the manuscript. We also admit the deficiencies in manuscript, as indicated by the Reviewer. In light of those comments, the following changes are made in the revised version, which are highlighted in text under turkis shading.
Reviewer#4
Comment 1: Line 97: Word reference error. Please Fix it
Comment 2: Line 102: Word reference error. Please Fix it
Comment 4: Line 226: Word reference error. Please Fix it
Comment 6: Line 230: Word reference error. Please Fix it
Response: The authors thank the reviewer for these carful observations. The auto-reference mistakes were corrected.
Comment 3: SEM micrographs of figure 3 and figure 5 are lacking resolution. Please improve the resolution and add higher magnification micrographs showing V(C,N).
Response: The authors appreciate the author remark and comment. The contrast and brightness of the SEM micrographs in Figure 3 and Figure 6 were adjusted to reveal the main microstructure features as recommended. However, the V(C,N) particles still need a high-resolution technique such as TEM to be clearly identified, which will be considered in the future work.
Comment 5: Line 229: typos error “…could be the reason behind the the higher martensite depths observed…”
Comment 9: Line 252: typos error “…rendes the pealitic phase transformation to proceed…”
Response: The authors do apologize for these typo-mistakes. They were corrected.
Comment 7: Line 233: how did you measure grain size? Explain in materials and methods section
Response: The grain size measurement method was added into the: 2. Materials and Methods section as follows:
The line intercept method was applied to measure the average grain size according to the guidelines of ASTM E-112 standard.
Comment 8: Table 3: martensite volume fraction is quite accurate; how did you measure it? Explain in Materials and Methods section
Response: The martensite volume fraction was calculated according to the stated formula below Table 3 (Mv (%) = [1-4(Rm/D)2]*100 ) and represented in Figure 4.
many thanks
Comment 8: Table 3: martensite volume fraction is quite accurate; how did you measure it? Explain in Materials and Methods section
Response: The martensite volume fraction was calculated according to the stated formula below Table 3 (Mv (%) = [1-4(Rm/D)2]*100 ) and represented in Figure 4.
Many thanks

Round 2
Reviewer 1 Report
The authors have made many revisions, making the manuscript more readable and the descriptions more accurate. Some explanation has been added for the major concern that direct observation of V-containing precipitations is absent, though not in a quite convincing way. However, much more care should be paid in preparing a satisfying manuscript, after which it can be considered for publication.
- Figure 2 in line 209 should be Figure 3.
- The light optical micrograph in line 111, 246, 260 are suggested to be modified to optical micrograph as the author did in line 213.
- Some references are missing, such as line 97, 102, 226, 230, and so on. More care should be made by the authors before submission.
- The two blue colors and two red colors in Figure 8 are too close to separate, except after enlarging quite a lot. They should be replaced with more distinguishable colors or characteristics.
- One of the conclusions the author added after revision that for steel grades with V-addition the productivity has been increased by applying higher rolling speeds seems confused, because both steel grades with and without V-addition were rolled at 13 and 7 m/s respectively. According to Table 2, different rolling speeds were designed just for different bar diameters.
There is no need for my further review if flaws mentioned above have been removed.
Author Response
Reviewer#1
The authors want to sincerely thank the Reviewer for his/her comments about the manuscript. We also admit the deficiencies in manuscript, as indicated by the Reviewer. In light of those comments, the following changes are made in the revised version, which are highlighted in text under turkis shading.
Comment 1: Figure 2 in line 209 should be Figure 3
Response: This mistake was corrected.
Comment 2: The light optical micrograph in line 111, 246, 260 are suggested to be modified to optical micrograph as the author did in line 213
Response: The authors followed the reviewer’s suggestion and modified the text accordingly.
Comment 3: Some references are missing, such as line 97, 102, 226, 230, and so on. More care should be made by the authors before submission.
Response: The authors do apologize for this auto-references error messages. Now all of these error messages were corrected.
Comment 4: The two blue colors and two red colors in Figure 8 are too close to separate, except after enlarging quite a lot. They should be replaced with more distinguishable colors or characteristics.
Response: We appreciate the reviewer’s comment and suggestion. Now the series in bar chart (figure 8) can be distinguished thanks to the reviewer’s observation
Comment 5: One of the conclusions the author added after revision that for steel grades with V-addition the productivity has been increased by applying higher rolling speeds seems confused, because both steel grades with and without V-addition were rolled at 13 and 7 m/s respectively. According to Table 2, different rolling speeds were designed just for different bar diameters.
Response: The authors do agree with reviewer’s opinion and they removed this conclusion to avoid the confusion.
